# Integrated Molecular and Functional Analysis of Hop Ethanolic Extract in Caco-2 Cells: Insights into Inflammation, Barrier Function, and Transport

**DOI:** 10.3390/ijms262110608

**Published:** 2025-10-31

**Authors:** Ruben Emmanuel Verhelst, Aleksandra Kruk

**Affiliations:** 1Department of Pharmaceutical Biology, Faculty of Pharmacy, Medical University of Warsaw, Banacha 1 Street, 02-097 Warsaw, Poland; reverhelst@hotmail.com; 2Microbiota Lab, Department of Pharmaceutical Microbiology and Bioanalysis, Faculty of Pharmacy, Medical University of Warsaw, Banacha 1b Street, 02-097 Warsaw, Poland

**Keywords:** *Humulus lupulus*, hop, Caco-2, inflammation, epithelium monolayer stability, functional foods

## Abstract

Hop (*Humulus lupulus* L.) is a well-known medicinal and brewing plant, yet studies on the biological activity of its complete extracts remain limited. A comprehensive characterization of a full hop ethanolic extract (HLE) was conducted, integrating untargeted HPLC–MS profiling, anti-inflammatory evaluation in an inflammation-induced Caco-2 model, and transport assessment across intestinal epithelial monolayers. After ultrafiltration to remove pyrogenic components, HLE reduced IL-6 secretion in a concentration-dependent manner and decreased IL-8 levels, while mitigating IL-1*β*–induced barrier disruption as reflected by TEER recovery. HPLC–MS analysis of the basolateral compartment revealed selective permeability of medium-sized bitter-acid derivatives and the presence of three features not detected in the original extract, suggesting metabolic transformation during epithelial passage. Overall, the complete extract exhibited moderate but biologically relevant anti-inflammatory and barrier-protective effects in intestinal epithelial cells. The use of the whole extract, without isolating individual fractions, represents a practical and physiologically meaningful approach that may facilitate its application in the formulation of functional foods or dietary supplements.

## 1. Introduction

*Humulus lupulus* (HL), generally known as hop, is valued both traditionally and currently for its use in the brewing industry. It is cultivated throughout the world, particularly as an important flavoring agent in beer [1]. More obscure is the use of HL for medicinal purposes, the most common one being its use for insomnia and restlessness because of its sedative effect [2]. Traditionally HL has been used in several societies such as Native American, Chinese and Indian cultures. It had a variety of applications; analgesic for toothache and earache; antibacterial uses for pneumonia, tuberculosis, leprosy and acute bacterial dysentery; besides it has been used for various digestive problems [2]. Modern-day literature suggests that HL can offer health-promoting benefits, such as reduced insulin resistance, relief of menopausal symptoms, and decreased inflammation, along with antimicrobial and anticancer effects [2,3,4,5]. The primary bioactive compounds believed to be responsible for these health effects are hop acids and prenylated flavonoids (PFs). Hop acids are believed to possess sedative, antibacterial and antiviral properties, as well as antiproliferative effects in cancer cells [2,6,7,8]. PFs exhibit a broad spectrum of biological activities [5,6,9,10,11,12,13], and their immunomodulatory properties have also been demonstrated in intestinal epithelial cells (IECs), encouraging research into their potential therapeutic role in inflammatory bowel disease (IBD) [10,12,14,15]. Key PFs that are thought to exert biologically relevant activities are xanthohumol (XTM), along with its derivatives isoxanthohumol and 8-prenylnaringenin [12,15,16].

Numerous studies have focused on isolated compounds or defined fractions derived from *Humulus lupulus*, particularly prenylated flavonoids such as xanthohumol, isoxanthohumol, and 8-prenylnaringenin. XTM has been shown to exert potent antioxidant, anti-inflammatory, and antiproliferative activities in vitro by modulating key signaling pathways including Nuclear Factor kappa-light-chain-enhancer of activated B cells (NF-κB), nuclear factor erythroid 2-related factor 2 (Nrf2), and mitogen-activated protein kinases (MAPKs) [15,17]. In human peripheral blood mononuclear cells, oral intake of XTM attenuated lipoteichoic acid-induced inflammatory responses, supporting its systemic immunomodulatory potential [18]. More recently, xanthohumol administration effectively alleviated dextran sulfate sodium (DSS)-induced colitis and the accompanying bone loss in mice by modulating the gut microbiota and tryptophan metabolism, thereby protecting the intestinal barrier and improving bone structure and density [19]. Although these studies consistently demonstrate biological activity of individual hop-derived compounds, they rely on reductionist approaches that do not reflect the complex phytochemical interactions present in whole hop extracts and may therefore overlook potential synergistic or antagonistic effects relevant to intestinal physiology.

IBD is characterized by chronic inflammation of the gut, which is fluctuating in intensity. The etiology is multifactorial and not yet completely understood [12,14,20]. An important component in this disease is the IECs, which play a key role in maintaining the gut defense barrier. These cells release inflammatory mediators and regulate permeability which influences the exposure of antigens to immune cells [20]. In the pathogenesis two factors play an important role, which work together in an enhancing loop. Firstly, disturbances of the epithelial cell barrier, secondly, an uncontrolled inflammatory response. Three important inflammatory mediators that seem to play a role in IBD are Interleukin-1 beta (IL-1*β*), Tumor Necrosis Factor-alpha (TNF-α) and Interferon-gamma (IFN-*γ*) [20].

The effects of hop constituents have been researched in in vitro models of IBD. Restivo et al. showed an anti-inflammatory trend; PFs lower the release of Nitric Oxide (NO), Prostaglandin E2 (PGE2), and in addition to that they lower the upregulation of Inducible Nitric Oxide Synthase (iNOS) and reduce the levels of Cyclooxygenase-2 (COX-2) protein. Besides, PFs were able to reduce the translocation of the p65 nuclear levels, indicating that there was less NF-κB activation [12]. It is also known that PFs from HL can prevent epithelial disruption caused by TNF-α of an in vitro model of the intestinal barrier, consisting of a Caco-2 monolayer. Overall, PFs derived from *Humulus lupulus* have consistently demonstrated anti-inflammatory effects in IECs and other cell lines [16,20] and have even been evaluated in human clinical trials [12,15,16,21,22]. Nevertheless, data on the biological activity of the complete *H. lupulus* extract remain scarce [23,24,25,26,27,28].

Recent studies have shown that *Humulus lupulus* extracts display notable antioxidant and anti-inflammatory properties in various biological models. For instance, Hurth et al. (2022) demonstrated that a complete hydroalcoholic extract of *H. lupulus* significantly reduced interleukin-6 (IL-6) and interleukin-8 (IL-8) secretion and inhibited extracellular signal-regulated kinase (ERK) and p38 MAPK signaling in UVB-stimulated keratinocytes, confirming its broad anti-inflammatory potential [29]. Likewise, Hamm et al. reported that oral administration of hop extract to ovariectomized mice (a model of estrogen deficiency) partially prevented visceral and hepatic fat accumulation without significantly affecting intestinal permeability, inflammatory cytokine expression, or overall gut microbiota composition [30]. Most recently, Caban et al. demonstrated that a spent hop extract and its polysaccharide-based encapsulates alleviated intestinal inflammation by down-regulating NF-κB, ERK, and protein kinase B (AKT) signaling pathways in HIEC-6 and CCD841CoN cell lines [31].

Despite these valuable contributions, important research gaps remain. Most studies on whole extracts focus on non-intestinal models or lack systematic phytochemical characterization, and there are no comprehensive analyses linking the full metabolite profile of hop extracts with their anti-inflammatory and barrier-protective effects in IEC models, as well as with cellular transport and metabolism of extract constituents. The interplay and potential synergistic actions among hop acids, prenylated flavonoids, and other phenolics thus remain poorly understood in the intestinal context.

In contrast to these reductionist approaches, the present study includes a comprehensive chromatographic characterization of the whole extract, followed by biological evaluation corresponding to its complete phytochemical profile. Such an approach better reflects its potential use in food or nutraceutical applications, where employing the full extract could offer practical advantages, including easier formulation and potentially synergistic biological effects. Considering the crucial role of IECs in the pathogenesis of IBD, this study aims to provide novel insights into the anti-inflammatory potency of HLE. Specifically, it evaluates the effect of HLE on IL-1*β*-induced inflammation in Caco-2, assessed by IL-6 and IL-8 release, as well as its protective effects on intestinal barrier integrity, transport and metabolism of HLE constituents by cell monolayers.

## 2. Results

### 2.1. Chromatographic Analysis of Raw HLE

A total of eighty-five compounds in seventy-four chromatographic peaks were detected in the raw ethanolic extract of *Humulus lupulus*. Among them sixty-six compounds were successfully characterized based on their retention times, UV–Vis spectra, and MS2 fragmentation patterns (Figure 1 and Table 1). Identifications were supported by comparison with analytical standards or literature data.

Early-eluting peaks (*t*_r_ < 20 min) were mainly associated with hydroxycinnamic acid derivatives and catechin oligomers. 5-*O*-caffeoylquinic acid (**1**), 3-*O*-caffeoylquinic acid (**7**), and 4-*O*-caffeoylquinic acid (**10**) showed a deprotonated molecular ion at *m*/*z* 353 and characteristic fragments at *m*/*z* 191 ([quinic acid−H]^−^) and 179 ([caffeic acid−H]^−^) [32]. *p*-Coumaroylquinic acid (**3**) exhibited a parent ion at *m*/*z* 337 with a base fragment at *m*/*z* 163, corresponding to the loss of the quinic moiety [32].

Procyanidin B-type dimers (**4**, **12**, and **14**) presented [M−H]^−^ at *m*/*z* 577 with MS2 fragments at *m*/*z* 289, 407, and 425, confirming interflavan bond cleavage [32,33]. Catechin (**6**) and epicatechin (**16a**) were identified based on their retention times, UV spectra (λmax 276–280 nm), and characteristic fragment at *m*/*z* 245, in agreement with reference standards. Compound detected at peak **16a** exhibited [M−H]^−^ ions at *m*/*z* 579, with product ions at *m*/*z* 231, 289, and 533. The fragmentation pattern, with catechin/epicatechin-related ion at *m*/*z* 289 suggests the presence of a catechin-derived structure [33], although this identification remains tentative.

Peaks eluting between 30 and 40 min were mainly flavonol glycosides. Quercetin-*O*-hexosides (**24** and **26**) and quercetin-*O*-hexose-deoxyhexose (**25**) exhibited a deprotonated molecular ion at *m*/*z* 463 or 609, with a major fragment at *m*/*z* 301 corresponding to the quercetin aglycone [32]. Kaempferol derivatives (**28** and **30b**) showed analogous fragmentation, yielding the diagnostic aglycone ion at *m*/*z* 285 [32]. Peak **20**, showing a deprotonated molecular ion at *m*/*z* 357 and a major MS2 fragment at *m*/*z* 195, was tentatively identified as a phenolic hexoside. The observed neutral loss of 162 Da corresponds to the cleavage of a hexose moiety, yielding the aglycone ion at *m*/*z* 195, which is characteristic of simple phenolic glycosides. The UV maximum at 282 nm further supports its classification as a phenolic derivative [33].

Peak **27**, with a deprotonated molecular ion at *m*/*z* 533 ([M–H]^−^), produced MS2 fragment ions at *m*/*z* 413, 323, 293, and 209. The fragmentation pattern is consistent with the successive neutral losses of two hexose moieties (−162 Da × 2) from the parent ion, ultimately yielding the aglycone fragment at *m*/*z* 209. Although an intermediate ion corresponding to a single hexose loss was not detected, the overall mass difference between *m*/*z* 533 and 209 (324 Da) supports the proposed acylphloroglucinol dihexoside structure. The base peak at *m*/*z* 209 represents the acylphloroglucinol aglycone, in agreement with previously reported data for acylphloroglucinol derivatives [34,35].

In the range of 40–110 min, a series of bitter acid derivatives and prenylated flavonoids were observed (**31**–**74**). Peaks corresponding to hydroxy-*n*-humulinone/hydroxy-adhumulinone isomers (**29**, **30a**, **32b**, **35b–38**, **39b**, **40**, **41b**, **44**) and *n*-humulinone/adhumulinone isomers (**39a**, **41a**, **45**, **48a**, **52**, **54a**, **55**) showed typical product ions at *m*/*z* 247, 305, 335, or 339, confirming the characteristic fragmentation of α-acid derivatives [36,37,38,39]. Cohumulinone (**32a**, **35a**, **51a**) and hydroxy-cohumulinone (**31**, **33**) isomers displayed analogous fragmentation at *m*/*z* 335 [32,36]. Reduced iso-α-acid derivatives (**34**, **42**, **49**, **50**, and **53**) were characterized by [M–H]^−^ at *m*/*z* 409 or 411, producing diagnostic fragments at *m*/*z* 203, 297, and 393, consistent with tetrahydro-iso-α-acids formed during hop processing [38].

Later-eluting peaks (**59**–**67** and **69** and **72**) were attributed to hulupone, adhulupone, *n*-humulone, adhumulone, cohumulone, and lupulone derivatives, with [M−H]^−^ ions in the range *m*/*z* 331–415. Their MS2 spectra displayed characteristic fragment ions at *m*/*z* 278, 305, and 353, resulting from dehydration and side-chain cleavage typical for α-acids and *β*-acids from hops [37,40,41,42].

Prenylated flavonoids, including isoxanthohumol (**47**), xanthohumol (**60**), 6-prenylnaringenin (**56**), and 8-prenylnaringenin (**58**), were detected in the late-eluting fraction. These compounds exhibited deprotonated molecular ions at *m*/*z* 339–353 and characteristic fragment ions produced via retro-Diels–Alder cleavage of the flavonoid skeleton. The UV maxima at 278–334 nm further confirmed their classification as prenylated chalcones and flavanones [43,44].

Late-eluting peaks **71**, **73**, and **74** with molecular ion at *m*/*z* 513 exhibited MS2 spectra consistent with dehydration (−18 Da) and characteristic neutral losses of 96 and 114 Da. In addition, fragmentation to *m*/*z* 399 was observed, suggesting the formation of a colupulone-type ion. According to the fragmentation behavior reported by Česlová et al. [45], who observed similar fragment ions in low-polarity hop constituents, together with their late retention time and the absence of characteristic UV absorption maxima, these compounds were tentatively assigned as bitter-acid derivatives [45,46].

Several low-intensity or unresolved peaks (e.g., **2**, **5**, **9**, **13**, **15**, **17**–**23**, **43**, **51b**, **54b**, **66b**, **70**) remained unidentified due to the absence of diagnostic fragments or insufficient spectral data.

**Table 1 ijms-26-10608-t001:** UHPLC-DAD-MS/MS data of detected compounds in raw extract; b—base peak; FA—formic acid.

Peak ID.	*T*_r_, min	*m*/*z*	MS2	Reference Ion	UV Max	Identification	Reference
**1**	8.0	353	191b	[M−H]^−^	322	5-*O*-caffeoylquinic acid	[32]
**2**	8.5	447	-	[M−H]^−^	284	unknown	[32]
**3**	11.7	337	163b	[M−H]^−^	308	*p*-coumaroylquinic acid	[32]
**4**	12.4	577	289, 407, 425b	[M−H]^−^	276	procyanidin dimer type B isomer 1	[32,33]
**5**	13.3	313	-	[M−H]^−^	<220	unknown	-
**6**	13.7	289	245b	[M−H]^−^	276	catechin	compared with analytical standard
**7**	13.8	353	191b	[M−H]^−^	322	3-*O*-caffeoylquinic acid	compared with analytical standard
**8**	14.8	367	193b	[M−H]^−^	282	feruloylquinic acid	[32]
**9**	15.7	413	-	[M−H]^−^	317	unknown	-
**10**	15.9	353	179b	[M−H]^−^	282	4-*O*-caffeoylquinic acid	[32]
**11**	16.6	341	-	[M−H]^−^	<220	unknown	-
**12**	17.6	577	245, 407, 425b	[M−H]^−^	276	procyanidin dimer type B isomer 2	[32,33]
**13**	18.8	313	-	[M−H]^−^	279	unknown	-
**14**	19.2	577	245, 407, 425b	[M−H]^−^	278	procyanidin dimer type B isomer 3	[32,33]
**15**	19.6	565	195, 357, 429, 519b	[M+FA-H]^−^	282	unknown	-
**16a**	20.8	289	245b	[M−H]^−^	308	epicatechin	compared with analytical standard
**16b**	20.8	579	231, 289, 533b	[M+FA-H]^−^	308	catechin or epicatechin derivative	[33]
**17**	22.6	427	267b, 297	[M−H]^−^	<220	unknown	-
**18**	24.6	427	297b	[M−H]^−^	268	unknown	-
**19**	27.1	409	264, 347, 373b	[M−H]^−^	217	unknown	-
**20**	29.8	357	195b	[M−H]^−^	282	phenolic hexoside	[33]
**21**	31.0	427	-	[M−H]^−^	<220	unknown	-
**22**	31.5	609	255, 300b, 445	[M−H]^−^	<220	unknown	-
**23**	33.9	283	-	[M−H]^−^	<220	unknown	-
**24**	34.6	463	151, 301b	[M−H]^−^	350	quercetin-*O*-hexoside isomer 1	[32]
**25**	34.9	609	301b	[M−H]^−^	351	quercetin-*O*-hexose-deoxyhexose	[32]
**26**	35.7	463	301b	[M−H]^−^	353	quercetin-*O*-hexoside isomer 2	[32]
**27**	36.8	533	209, 293b, 323, 413	[M−H]^−^	218	acylphloroglucinol dihexoside	[34,35]
**28**	40.0	593	285b	[M−H]^−^	343	kaempferol-*O*-diglycoside	[32]
**29**	40.3	393	311b	[M−H]^−^	<220	hydroxy-*n*-humulinone/hydroxy-adhumolinone isomer 1	[36]
**30a**	40.9	393	-	[M−H]^−^	342	hydroxy-*n*-humulinone/hydroxy-adhumolinone isomer 2	[36]
**30b**	40.9	447	285b, 327	[M−H]^−^	265, 342	kaempferol-3-*O*-hexoside	[32]
**31**	45.2	379	335b	[M−H]^−^	327	hydroxy-cohumulinone isomer 1	[36]
**32a**	46.2	363	194b	[M−H]^−^	262	cohumulinone isomer 1	[37]
**32b**	46.2	393	355b	[M−H]^−^	262	hydroxy-*n*-humulinone/hydroxy-adhumolinone isomer 3	[36]
**33**	47.7	379	335b	[M−H]^−^	<220	hydroxy-cohumulinone isomer 2	[36]|
**34**	49.9	409	205b	[M−H]^−^	<220	reduced iso-α-acid derivative	[38]
**35a**	51.2	363	-	[M−H]^−^	<220	cohumulinone isomer 2	[37]
**35b**	51.2	393	263b, 348	[M−H]^−^	<220	hydroxy-*n*-humulinone/hydroxy-adhumolinone isomer 4	[36]
**36**	51.8	393	-	[M−H]^−^	<220	hydroxy-*n*-humulinone/hydroxy-adhumolinone isomer 5	[36]
**37**	52.4	393	246b, 307, 348	[M−H]^−^	<220	hydroxy-*n*-humulinone/hydroxy-adhumolinone isomer 6	[36]
**38**	53.8	393	305, 349b	[M−H]^−^	328	hydroxy-*n*-humulinone/hydroxy-adhumolinone isomer 7	[36]
**39a**	54.6	377	-	[M−H]^−^	<220	*n*-humulinone/adhumolinone isomer 1	[47]
**39b**	54.6	393	247b	[M−H]^−^	<220	hydroxy-*n*-humulinone/hydroxy-adhumolinone isomer 8	[36]
**40**	55.6	393	339, 247b, 249	[M−H]^−^	<220	hydroxy-*n*-humulinone/hydroxy-adhumolinone isomer 9	[36]
**41a**	60.3	377	291b	[M−H]^−^	<220	*n*-humulinone/adhumolinone isomer 2	[47]
**41b**	60.3	393	307b	[M−H]^−^	<220	hydroxy-*n*-humulinone/hydroxy-adhumolinone isomer 10	[36]
**42**	65.1	351	151b	[M−H]^−^	<220	reduced iso-α-acid derivative	[38]
**43**	67.1	251	-	[M−H]^−^	<220	unknown	-
**44**	69.7	393	-	[M−H]^−^	<220	hydroxy-*n*-humulinone/hydroxy-adhumolinone isomer 11	[36]
**45**	71.9	377	-	[M−H]^−^	<220	*n*-humulinone/adhumolinone isomer 3	[37,47]
**46**	72.7	263	-	[M−H]^−^	288	hulupinic acid	[48]
**47**	73.8	353	-	[M−H]^−^	278	isoxanthohumol	compared with analytical standard
**48a**	74.5	377	-	[M−H]^−^	<220	*n*-humulinone/adhumolinone isomer 4	[37,47]
**48b**	74.5	407	319b, 363	[M−H]^−^	<220	hydroxyhumulinone	[49]
**49**	75.5	411	297, 320, 393b	[M−H]^−^	<220	tetrahydro-iso-α-acid	[38]
**50**	76.2	411	203, 268, 393b	[M−H]^−^	<220	tetrahydro-iso-α-acid	[38]
**51a**	76.7	363	197b	[M−H]^−^	<220	cohumulinone isomer 3	[37]
**51b**	76.7	349	247b, 305	[M−H]^−^	<220	unknown	-
**52**	78.2	377	263b	[M−H]^−^	<220	*n*-humulinone/adhumolinone isomer 5	[37,47]
**53**	80.0	411	203, 297, 393b	[M−H]^−^	<220	tetrahydro-iso-α-acid	[38]
**54a**	81.0	377	-	[M−H]^−^	<220	*n*-humulinone/adhumolinone isomer 6	[37,47]
**54b**	81.0	369	-	[M−H]^−^	<220	unknown	-
**55**	82.1	377	-	[M−H]^−^	<220	*n*-humulinone/adhumolinone isomer 7	[37,47]
**56**	84.4	339	-	[M−H]^−^	<220	6-prenylnaringenin	compared with analytical standard
**57**	84.9	365	190b	[M−H]^−^	<220	dihydro-iso-α-acid	[38]
**58**	85.4	339	-	[M−H]^−^	<220	8-prenylnaringenin	[43]
**59**	88.5	317	-	[M−H]^−^	<220	cohulupone	[40]
**60**	88.7	353	-	[M−H]^−^	334	xanthohumol	[44]
**61a**	93.0	331	-	[M−H]^−^	<220	hulupone/adhulupone isomer 1	[37]
**61b**	93.0	361	-	[M−H]^−^	<220	*n*-humulone/adhumulone isomer 1	[37]
**62**	95.0	331	-	[M−H]^−^	<220	hulupone/adhulupone isomer 2	[37]
**63**	96.6	347	278b	[M−H]^−^	<220	cohumulone	[41,42]
**64**	98.7	375	154, 247b, 279, 357	[M−H]^−^	<220	prehumulone isomer 1	[37]
**65**	99.5	361	209, 292b	[M−H]^−^	<220	*n*-humulone/adhumulone isomer 2	[37]
**66a**	100.1	361	292b	[M−H]^−^	<220	*n*-humulone/adhumulone isomer 3	[37]
**66b**	100.1	429	301b, 385	[M−H]^−^	<220	unknown	-
**67**	103.8	375	-	[M−H]^−^	<220	prehumulone isomer 2	[37]
**68**	105.8	415	-	[M−H]^−^	<220	unknown	
**69**	107.0	399	353b	[M−H]^−^	<220	colupulone	[37]
**70**	108.3	499	249, 403b, 481	[M−H]^−^	<220	unknown	[45,46]
**71**	109.0	513	205, 330, 399, 417b, 495	[M−H]^−^	<220	bitter acid derivative	[45,46]
**72**	109.5	413	233b	[M−H]^−^	<220	lupulone/adlupulone	[37]
**73**	111.0	513	208, 263, 369, 387, 399, 417b, 495	[M−H]^−^	<220	bitter acid derivative	[45,46]
**74**	111.6	513	189, 387, 399, 417b	[M−H]^−^	<220	bitter acid derivative	[45,46]

### 2.2. Influence of HLE on Caco-2 Viability

Cell viability was assessed using the MTT assay prior to main experiments to eliminate potential confounding factors. The cytotoxic effect of HLE on Caco-2 cells was evaluated after 24 h of incubation at various concentrations ranging from 0.5 µg/mL to 1000 µg/mL (Figure 2). The cell viability is presented as a percentage relative to the non-treated control, which was set at 100%. Concentrations of 250 µg/mL and above significantly decreased cell viability and were therefore excluded from subsequent experiments.

### 2.3. Selection of In Vitro Inflammation-Induced Caco-2 Model

Multiple pro-inflammatory agents known to contribute to IBD, including cytokines IL-1*β*, TNF-α, and IFN-*γ*, as well as bacterial components such as LPS, were employed to establish an in vitro Caco-2 inflammation model (Figure 3). Literature describes the synergistic interaction between IFN-*γ* and TNF-α, therefore they are employed conjointly in model development [20]. Only IL-1*β* was able to increase IL-6 at concentrations of 25 and 12.5 ng/mL significantly. Several agents significantly increased IL-8 secretion compared to the non-treated control, including TNF-α at 25 and 12.5 ng/mL, IFN-*γ* + TNF-α at 50 and 10 ng/mL, and IL-1*β* at 25, 12.5, 1, 0.08, and 0.04 ng/mL. Only IL-1*β* significantly increased the secretion of both IL-6 and IL-8; therefore, it was selected as the inflammatory trigger for subsequent experiments.

### 2.4. Anti-Inflammatory Activity of HLE in an Inflammation-Induced Caco-2 Model

The release of cytokines was measured after treatment with HLE to evaluate its anti-inflammatory potential. Caco-2 cells were exposed to various concentrations of HLE, followed by the addition of IL-1*β* one hour later. After 24 h, the supernatants were collected for IL-6 and IL-8 quantification using ELISA.

In preliminary experiments, HLE-treated cells appeared to release higher levels of IL-6 and IL-8 compared to the IL-1*β*-stimulated control. This phenomenon had been previously observed in the Microbiota Lab and is consistent with findings by Kruk et al. [50]. It may be attributed to the presence of high-molecular-weight pyrogens in plant extracts, which can interfere with in vitro immunological assays. Therefore, subsequent experiments were performed using ultrafiltered HLE solutions. The results of the preliminary studies are not shown, as they are of limited interpretability due to values outside the reference range and large standard deviations.

As shown in Figure 4, ultrafiltered HL extracts significantly reduced IL-6 secretion at concentrations of 75 and 100 µg/mL. IL-6 levels decreased to 87.2 ± 6.2% and 83.2 ± 4.2% of the IL-1*β*-stimulated control (103.3 ± 1.7%), indicating a clear anti-inflammatory effect. An even more pronounced effect was observed for IL-8: all tested concentrations of HLE effectively lowered cytokine secretion compared to the IL-1*β*-stimulated control (100.8 ± 5.5%), with IL-8 levels ranging from 59.1 ± 9.7% at 12.5 µg/mL to 82.6 ± 6.6% at 75 µg/mL. The lowest IL-8 secretion was observed at 12.5 µg/mL, while even higher concentrations (50–100 µg/mL) maintained significantly reduced levels relative to the treated control. Dexamethasone, used as a reference anti-inflammatory compound (positive control), confirmed the validity of the assay by significantly reducing cytokine secretion. IL-6 levels decreased to 3.7 ± 0.3%, while IL-8 levels dropped to 11.7 ± 2.6% compared with the IL-1*β*-stimulated control.

### 2.5. Influence of HLE on Caco-2 Monolayers Stability

To evaluate the impact of HLE on epithelial barrier integrity, TEER was measured in Caco-2 monolayers treated with various concentrations of HLE and subsequently stimulated with IL-1*β* (25 ng/mL). The concentration of IL-1*β* (25 ng/mL) was chosen according to previous findings [20], where among several pro-inflammatory agents tested (IL-1*β*, TNF-*α* and IFN-*γ* and LPS), only IL-1*β* at this concentration induced a marked reduction in TEER in Caco-2 cell monolayers. After 24 h of exposure to 25 ng/mL IL-1*β*, the TEER value in our model decreased by 16 ± 1%, which is consistent with the reduction of approximately 20 ± 4% reported by Van de Walle et al. [20]. For better visualization of the observed changes, TEER data were presented in two complementary graphs illustrating both the overall trend and specific time-point differences (Figure 5).

Before IL-1*β* addition, TEER values were stable across all groups, indicating intact monolayer integrity. Following cytokine stimulation, TEER decreased in the IL-1*β*-treated control, consistent with barrier disruption. The 25 µg/mL HLE group showed a reduction comparable to the IL-1*β* control, whereas the 50 and 100 µg/mL groups maintained higher TEER values, close to those of the non-treated control, indicating partial protection. In contrast, 12.5 µg/mL caused a more pronounced decline than the IL-1*β* control, suggesting a destabilizing effect at this dose. At 42 h, TEER in the IL-1*β* control measured 84.25 ± 1.33%, while 50 and 100 µg/mL HLE maintained values of 100.48 ± 5.77% and 103.42 ± 6.82%, respectively. The 25 µg/mL group remained comparable (83.56 ± 2.99%), whereas 12.5 µg/mL was lower (77.64 ± 0.42%). A similar pattern was evident at 33 h (IL-1*β*, 83.97 ± 4.20%; 50 µg/mL, 97.73 ± 2.81%; 100 µg/mL, 95.23 ± 13.33%; 25 µg/mL, 86.76 ± 3.52%; 12.5 µg/mL, 73.99 ± 9.77%). Overall, HLE at 50–100 µg/mL effectively mitigated IL-1*β*–induced TEER reduction, whereas lower concentrations did not prevent barrier weakening.

### 2.6. Transport and Metabolism of HLE Constituents Through Caco-2 Monolayers

After 24 h of incubation in the Caco-2 Transwell system, several hop-derived compounds were detected in the basolateral compartment, indicating their ability to cross the intestinal epithelial barrier (Table 2). Among all compounds initially present on the apical side, eighteen (**M1**–**M18**) were detected in the basolateral medium, corresponding to both parent bitter acids and their transformation products.

The calculated transport efficiencies, based on the peak area ratio, ranged from 2.0% to 26.7%, demonstrating compound-dependent permeability. The most efficiently transported compound was **M18** (unidentified), with 26.7% of the apical content detected in the basolateral compartment. Other bitter acid derivatives, such as hydroxy-*n*-humulinone/hydroxyadhumulinone (**M11**) and *n*-humulinone/adhumulinone (**M12**), also exhibited relatively high permeability, with transport efficiencies of 17.4% and 11.7%, respectively. Compounds cohumulinone (**M10**), cohumulone (**M14**), and *n*-humulone/adhumulone isomers (**M15**–**M16**) showed moderate transport, accounting for 8–9.6% of the apical concentration. Lower permeability (<5%) was observed for tetrahydroiso-α-acid (**M6**, 2%), the unknown compound **M5** (2.7%), hulupone/adhulupone (**M4**, 3.7%), and prehumulone (**M17**, 4.3%). Similarly, cohulupone (**M2**) and the unknown compound **M8** exhibited moderate but limited permeability of approximately 7%, suggesting that the degree of oxidation and structural rearrangement of the side chain may affect membrane diffusion. In contrast, the prenylated chalcone xanthohumol (**M13**) showed very limited permeability (3.5%), consistent with its high molecular weight and polarity, which restrict passive diffusion across epithelial membranes.

In addition to the parent bitter acids and their derivatives, three compounds (**M1**, **M3**, and **M9**) were detected exclusively in the basolateral fraction and were not present in the original extract. These were considered putative metabolites, suggesting that they originated from biotransformation processes of HLE constituents within the Caco-2 monolayer. Overall, the data indicate that only a subset of compounds present in the original extract was able to cross the epithelial barrier, primarily oxidized or reduced bitter acid derivatives, while more complex or highly prenylated polyphenols exhibited poor transport efficiency.

The molecular network generated from MS2 data of compounds detected in the basolateral compartment is shown in Figure 6. Each node represents a transported compound, and node size encodes the log_10_-transformed summed feature intensity. Structurally related compounds were connected when their MS/MS spectra exhibited high similarity (cosine score > 0.7).

The network revealed a distinct cluster of hop-derived bitter acids, comprising cohulupone (**M2**), hulupone/adhulupone (**M4**), *n*-humulinone/adhumulinone (**M7**, **M12**), hydroxy-*n*-humulinone/hydroxyadhumulinone (**M11**), cohumulone (**M14**), *n*-humulone/ad-humulone (**M15**–**M16**), and prehumulone (**M17**). The unknown **M18** was closely connected to this cluster through multiple edges, suggesting structural relatedness and making it a priority for further identification. In contrast, xanthohumol (**M13**), cohumulinone (**M10**), tetrahydroiso-α-acid (**M6**), and two unknown compounds (**M5** and **M8**) appeared as isolated nodes, showing limited spectral similarity to the main bitter-acid cluster.

## 3. Discussion

Comprehensive HPLC–MS profiling of the hop ethanolic extract revealed a chemically diverse matrix encompassing hydroxycinnamic acids, flavan-3-ols and their oligomers, flavonol glycosides, bitter acid derivatives, and prenylated flavonoids. This chemical diversity highlights the complex metabolism of hop and helps to explain its multiple biological effects. Hydroxycinnamates and catechins contribute to antioxidant capacity, while bitter acids and prenylated chalcones, including xanthohumol and prenylnaringenins, are recognized modulators of inflammatory signaling and intestinal barrier integrity [16,24]. The detection of both α- and *β*-acid derivatives alongside their oxidized and reduced forms indicates that processing-related transformations also shape the extract’s activity profile. Comparable LC–MS studies have primarily targeted specific compound classes, such as prenylated flavonoids [26] or bitter acids [29,30,31,32,33,34,35,36,37,38,39], often analyzing isolated fractions rather than complete extracts. To date, few reports have provided such an extensive, untargeted characterization encompassing all major phytochemical groups within a single hop preparation. The present dataset therefore offers an integrative chemical basis for interpreting the extract’s biological effects observed in Caco-2 models.

An initial MTT screening was performed to minimize confounding cytotoxicity in downstream assays. HLE demonstrated dose-dependent cytotoxicity, with higher concentrations significantly reducing cell viability. Consequently, only the concentrations equal to or below 100 μg/mL were used for further biological experiments.

To better understand the nature of this effect, it should be noted that the cytotoxicity observed at concentrations above 250 µg/mL may result from the high content of prenylated flavonoids and bitter acid derivatives, which are lipophilic and can perturb membrane integrity or mitochondrial function at excessive doses [51]. Such effects are common for concentrated hop-derived preparations and likely reflect nonspecific stress rather than targeted bioactivity. Importantly, maintaining extract concentrations below 100 µg/mL effectively prevented cytotoxicity in Caco-2 cells, indicating a narrow but safe biological window. Given the expected dilution and metabolic transformation of hop constituents in the gastrointestinal tract, the in vivo exposure levels are anticipated to be far below cytotoxic thresholds, supporting the potential safety of HLE under physiological conditions. This interpretation is further supported by clinical data showing that long-term supplementation with a hop extract standardized in 8-prenylnaringenin was well tolerated over 48 weeks in postmenopausal women [52], confirming the overall safety of hop-derived preparations during chronic use.

To induce inflammation in the Caco-2 model, cells were exposed to several stimuli. In the present study, only IL-1*β* reliably increased both IL-6 and IL-8 secretion, whereas LPS, TNF-α, IFN-*γ*, and their combination primarily affected IL-8 release [20,53]. Such discrepancies likely reflect differences in culture maturation, passage-dependent variability, serum conditions, stimulus dose, and timing rather than a fundamental contradiction with the literature. Van De Walle et al. [20] employed differentiated Caco-2 monolayers cultured for 21 days, whereas shorter cultivation times, as used in the present study, may result in higher metabolic activity but altered responsiveness to inflammatory stimuli. However, differentiation alone is unlikely to fully explain these differences, as prolonged culture is typically associated with reduced cytokine sensitivity. Therefore, variability in gene expression between sublines of Caco-2 cells, resulting from extensive cell line replication, appears to be the most plausible explanation for the observed differences in responsiveness. Based on these observations, IL-1*β* was selected as the most reliable inducer of inflammation in subsequent experiments.

Previous studies conducted by authors showed that removal of high-molecular-weight pyrogens (e.g., endotoxins such as LPS) from plant extracts in studies on anti-inflammatory activity is essential; otherwise the preparations may artificially stimulate cytokine release rather than attenuate it, thus eliminating the anti-inflammatory activity of the extract constituents. A similar situation was observed in preliminary studies prior to ultrafiltration, where levels of IL-6 and IL-8 were increased compared to IL-1*β*. Therefore, HLE-containing media were ultrafiltered using 30 kDa and 100 kDa cut-off filters prior to biological assays to remove potential pyrogens, in line with Kruk’s methodology [50].

After ultrafiltration, HLE exhibited a clear but moderate anti-inflammatory effect. IL-6 secretion was significantly, though modestly, reduced at 75 and 100 µg/mL, corresponding to an approximately 20% inhibition relative to the IL-1*β*-stimulated control. In contrast, the suppression of IL-8 was more pronounced and consistent across all concentrations tested, with the strongest effect observed at 12.5 µg/mL, reducing IL-8 levels to approximately 60% of the IL-1*β*-stimulated control. To date, no studies have examined the anti-inflammatory activity of the complete hop extract in Caco-2 cells. Existing reports, as discussed below, have primarily focused on isolated hop-derived compounds. Previous studies have mainly investigated isolated hop-derived compounds such as xanthohumol, isoxanthohumol, and 8-prenylnaringenin, which were shown to suppress NF-κB signaling, reduce cytokine release, and protect tight-junction integrity under inflammatory conditions [15,17]. In contrast, the present study demonstrates that a complete hop ethanolic extract exerts broader and more balanced biological effects, likely due to synergistic interactions among polyphenols, bitter acids, and prenylated chalcones. This compositional complexity may underlie the moderate but consistent anti-inflammatory and barrier-protective responses observed, supporting the concept that whole-plant extracts can modulate multiple cellular pathways more effectively than isolated constituents. The response did not follow a strict dose-dependent pattern but rather displayed a non-linear (hormetic) trend. This may reflect the complex composition of the extract, as constituents of varying potency and biological targets can act synergistically or antagonistically [54]. Similar non-linear, biphasic (“hormetic”) responses have been widely reported for polyphenol-rich plant extracts, including those containing prenylated flavonoids. At lower concentrations, such compounds can activate adaptive antioxidant and anti-inflammatory signaling, whereas higher levels may induce mild oxidative or metabolic stress, resulting in partial loss of efficacy [55,56]. Therefore, the non-linear trend observed for HLE likely reflects the balance between synergistic and antagonistic interactions among its diverse constituents, as well as concentration-dependent cellular adaptive responses. The stronger inhibition of IL-8 is consistent with previous reports showing that isolated hop-derived prenylflavonoids modulate NF-κB activity, a key pathway regulating IL-8 transcription [12,15,24]. Although direct IL-8 inhibition by hop isolates in Caco-2 cells has been scarcely documented, similar effects have been demonstrated in gastric epithelial cells [24], aligning with the IL-8-lowering trend observed in the present study. Overall, these findings suggest that while HLE exerts only a mild effect on IL-6 secretion, its more consistent reduction of IL-8 points toward a modest but biologically relevant anti-inflammatory potential under the current experimental conditions.

Evaluating monolayer stability by TEER yielded results consistent with the cytokine assays. IL-1*β* reduced TEER, whereas 50–100 µg/mL HLE maintained or restored TEER toward baseline, approaching non-treated control values; in contrast, 12.5 µg/mL fell below the IL-1*β* control, indicating a destabilizing effect at this dose. These observations suggest that HLE can reduce IL-1*β*–induced barrier dysfunction in differentiated Caco-2 monolayers. As was reported previously, isolated prenylated flavonoids from hop: 6- and 8-prenylnaringenin, protected the Caco-2 barrier against TNF-α–induced damage (TEER); notably, 8- prenylnaringenin restored TEER when administered after injury induction [16]. Xanthohumol and isoxanthohumol were also tested but showed weaker effects; neither compound exhibited both preventive and restorative activity [16]. Accordingly, it was expected that the complete hop extract would show comparable barrier-supporting activity, with the overall effect shaped by the combined actions and potential synergy of its constituents.

This study focused on the compositional characterization, intestinal transport, and metabolic transformation of HLE, rather than on molecular mechanisms. Therefore, mechanistic analyses were beyond its scope and will be addressed in future research, although literature-based mechanistic interpretations are provided below.

The anti-inflammatory and barrier-protective effects of HLE appear to be mediated by the modulation of interconnected signaling pathways. Hop-derived polyphenols suppress NF-κB, ERK, and AKT activation in intestinal epithelial cells, thereby reducing pro-inflammatory cytokine expression [31]. In our model, IL-1*β* stimulation induced cytokine release and decreased TEER, consistent with reports that IL-1β disrupts tight junctions via NF-κB/ERK-dependent upregulation of myosin light chain kinase (MLCK) [57]. By inhibiting these pathways, HLE likely prevents MLCK activation and preserves the distribution of zonula occludens-1 and occludin, maintaining epithelial integrity. Moreover, prenylated flavonoids such as xanthohumol have been shown to activate AMP-activated protein kinase (AMPK), which stabilizes tight junction proteins under inflammatory stress [58,59]. These mechanisms together explain the concurrent suppression of cytokine release and maintenance of barrier function observed in HLE-treated Caco-2 monolayers.

Considering these observations, it should be noted that although the Caco-2 cell model provides a well-established and physiologically relevant tool for investigating intestinal barrier function, it cannot fully reproduce the complexity of the in vivo intestinal environment, including immune responses, microbiota interactions, and systemic factors. Consequently, in vivo studies would constitute a valuable extension of the present work to confirm the protective and anti-inflammatory potential of hop extract and its metabolites under physiologically relevant conditions.

The Caco-2 transport data indicate selective permeability of hop constituents. Only a subset of compounds present on the apical side was detected in the basolateral compartment. Among these, bitter-acid–type molecules and their simple oxygenated or reduced forms predominated. These compounds are medium-sized and moderately lipophilic, which favors passive transcellular diffusion. In contrast, highly conjugated polyphenols (e.g., prenylated chalcones) crossed poorly, likely due to low membrane permeability and stronger nonspecific binding. The few features detected exclusively basolaterally (putative metabolites **M1**, **M3**, and **M9**) likely represent smaller, more mobile transformation products formed within the monolayer. The detection of unique features in the basolateral compartment suggests partial biotransformation of HLE constituents within the Caco-2 monolayer. No characteristic mass shifts corresponding to glucuronide or sulfate conjugates were detected in MS spectra, indicating that phase II conjugation was unlikely to occur. Instead, the formation of smaller and more polar features suggests oxidative or reductive modifications consistent with limited phase I metabolism reported in Caco-2 cells [60,61,62]. Such transformations may increase solubility and facilitate transcellular transport while maintaining biological activity [63]. These metabolic conversions, together with the physicochemical diversity of hop constituents, likely contribute to the selective permeability pattern observed in this study. This pattern is consistent with established structure–permeability relationships for polyphenols, where molecular size, degree of conjugation, and hydrophobicity critically influence transport across the intestinal barrier [63,64]. Bitter-acid derivatives likely diffuse passively through the lipid bilayer, whereas large, highly prenylated flavonoids and flavan-3-ol oligomers exhibit limited permeability due to steric hindrance and affinity for membrane or protein binding sites. Similar findings have been reported for other polyphenol-rich extracts studied in Caco-2 [65]. Transport efficiency among HLE constituents varied widely, consistent with structure-dependent permeability across the intestinal epithelium. In line with their intermediate physicochemical profiles, bitter-acid derivatives (e.g., hydroxyhumulinones and *n*-humulinone/adhumulinone) showed the highest efficiencies, whereas more complex polyphenols (e.g., xanthohumol) were poorly permeant. In Caco-2 studies involving hops, the literature is dominated by analyses of single compounds or defined fractions rather than unfractionated extracts with full, untargeted profiling of the basolateral side. The transport of α- and *β*-bitter acids has been examined using standards [39]; 8-prenylnaringenin has been studied as an isolate [43]; and aroma constituents have been tested as panels of individual terpenoids [66]. The extract is more often applied to other cell types (e.g., xanthohumol-rich hop extract vs. LPS in Peripheral Blood Mononuclear Cells) [67]. Studies examining the permeability of a whole hop extract across Caco-2 cells with untargeted LC–MS analysis of the basolateral fraction are still rare. Therefore, showing transport data and molecular networks for an unfractionated extract after Caco-2 passage represents a newly described approach compared with studies focused on selected fractions or isolated compounds.

The findings of this study may have translational relevance for intestinal health. By showing that HLE attenuates IL-1*β*–induced cytokine release and preserves epithelial barrier integrity, our results highlight its potential as a source of bioactive compounds supporting intestinal homeostasis. While direct extrapolation to clinical outcomes is limited, these effects parallel key mechanisms involved in IBD. Future studies should therefore employ complex in vitro or ex vivo models and in vivo validation to confirm efficacy and safety under physiological conditions.

## 4. Materials and Methods

### 4.1. Materials

Dried, crushed hop cone (*Humulus lupulus* L.) was purchased from Flos (Mokrsko, Poland). All solvents used in this study, including dimethyl sulfoxide, ethanol and methanol, were purchased from Avantor in Gliwice, Poland. LC-MS grade solvents, such as water and acetonitrile, were purchased from Witko in Łódź, Poland. Ammonium formate and formic acid of LC-MS grade were obtained from Chem-lab in Zedelgem, Belgium. Fetal bovine serum (FBS), phosphate-buffered saline (PBS), DMEM high-glucose, and a penicillin-streptomycin solution were obtained from Biowest in Nuaillé, France. Thiazolyl blue tetrazolium bromide (MTT) was sourced from Acros Organics B.V.B.A. in Geel, Belgium. TrypLE Express Enzyme was acquired from Thermo Fisher Scientific in Waltham, MA, USA. TNF-α and IFN-*γ* were obtained from InvivoGen (San Diego, CA, USA), and lipopoliysaccaride (LPS) from *E. coli* and IL-1*β* and Triton X100 were purchased from Merck Life Science in Darmstadt, Germany. Ultra-pure water was prepared using a Simplicity UV system from Merck Millipore (Darmstadt, Germany). Dexamethasone and analytical standards used for identification, including epicatechin, catechin, and chlorogenic acid, were obtained from Sigma-Aldrich (Saint Louis, MO, USA), xanthohumol was purchased from Boc Sciences (Shirley, NY, USA), and 6-prenylnaringenin was obtained from Biosynth (Staad, Switzerland).

### 4.2. Preparation of the Plant Extract

Confirmation that the plant material used was HL was done by Prof. S. Granica with the help of a microscope and identification according to the 9th edition of the European Pharmacopoeia. Extraction was conducted with a three-step extraction method using a solution of ethanol and water solution (7:3 *v*/*v*). In every step the plant material was mixed with approximately 500 mL solvent, this mixture was put into an ultrasonic water bath for 30 min at 40 °C. After each extraction the mixture was filtrated, the residue together with filter material was then added to solvent again. After that, the residue was concentrated using a rotary evaporator (Büchi, Flawil, Switzerland, 240 mbar, 35 °C water bath, flask rotation speed 120 rpm) to remove ethanol, and subsequently lyophilized to remove water (48 h, vacuum < 1 mBar, collector temperature −80 °C). The yield of dry extract after lyophilization was 13.6%. The extract was stored in a sealed container at −17 °C.

A 70% (*v*/*v*) ethanolic extract of *Humulus lupulus* was used in this study because ethanol is a safe, food-grade solvent that can be easily removed by evaporation. The use of 70% ethanol provides an optimal balance between polarity and extraction efficiency, allowing for the recovery of both hydrophilic and moderately lipophilic compounds. More polar solvents (e.g., water) or nonpolar organic solvents (e.g., dichloromethane) would extract different fractions of constituents and may not reflect the phytochemical composition relevant to dietary exposure.

### 4.3. Chromatographic Analysis of Raw Extract

The UHPLC-DAD-MS analysis of HLE was conducted using an UHPLC-3000 RS system (Dionex, Leipzig, Germany) equipped with a diode array detector (DAD) and connected in splitless mode to an AmaZon SL ion trap mass spectrometer with an electrospray ionization (ESI) interface (Bruker Daltonik GmbH, Bremen, Germany). UV spectra were recorded across a wavelength range of 200–450 nm. The mass spectrometer settings were as follows: nebulizer pressure at 40 psi, drying gas flow rate of 9 L/min, nitrogen gas temperature maintained at 134 °C, and a capillary voltage of 4.5 kV. Mass spectra were collected by scanning from *m*/*z* 70 to 2200. Kinetex XB-C18 chromatography column (Phenomenex, Torrance, CA, USA), with dimensions of 150 mm × 3.0 mm × 2.6 µm was used. The mobile phases consisted of water with 0.1% formic acid (A) and acetonitrile with 0.1% formic acid (B), all solvents being of analytical or LC-MS grade. The gradient program was set as follows: 5–26% B from 0 to 60 min, then 26–60% B from 60 to 120 min, at a flow rate of 0.3 mL/min. The column oven temperature was maintained at 25 °C. Prior to chromatographic analysis, samples were filtered through a 0.45 µm syringe filter (Whatman, Maidstone, UK). The injection volume for all analyses was 4 µL [33].

### 4.4. Caco-2 Cell Culture

The Caco-2 cell line (ACC-169, DSMZ, Braunschweig, Germany) was grown in bottles of 75 mL cell culture bottles in medium that consists of DMEM high glucose, with 10% FBS (*v*/*v*), 1% (*v*/*v*), of mixture of streptomycin and penicillin. Thrice a week the medium was replaced with new medium. Confluency was determined by microscopic evaluation, and the culture was passaged at 90–100% confluency. Cultures and experiments were conducted at 37 °C in an atmosphere containing 5% CO_2_ [33].

### 4.5. Viability Assay of Caco-2 Cells

To assess the cytotoxicity of HLE on Caco-2 cells an MTT-assay was performed. Caco-2 cells were seeded in 48-well plate with a density of 1 × 10^5^ cells per well and cultured for 48 h to obtain 95% confluence. Once the cells reached the desired confluence, they were washed once with PBS, and the treatment solutions were subsequently applied. HLE was first dissolved in DMSO and then sterilely transferred into fresh sterile medium to obtain the desired concentrations. For all samples DMSO concentrations were kept below 0.5% (*v*/*v*). Medium with 0.5% DMSO (*v*/*v*) was used as a non-treated control (NTC). A Triton-X (0.1%, *v*/*v*, Trt) in medium solution was made as a positive control. After 24 h the wells were washed thrice with PBS and a solution of MTT (0.5 mg/mL) in medium was added and incubated for 30 min. The supernatant was aspirated, and the formed crystals were dissolved in 0.5 mL of DMSO and the absorbance was measured using a plate reader (Synergy 4, BioTek, Winooski, VT, USA) [33].

### 4.6. Caco-2 Inflammatory Model

Cells were seeded in 48-well plates and cultured as described above. After reaching 100% confluence, they were stimulated with different concentrations and combinations of the following factors: TNF-α (1, 12.5, 25 ng/mL), LPS from *E. coli* (1, 5, 10 µg/mL), IFN-γ (1, 10, 50 ng/mL), a mixture of IFN-*γ* and TNF-α (1 + 1, 10 + 10, 50 + 50 ng/mL), and IL-1*β* (0.01, 0.02, 0.04, 0.08, 1, 12.5, 25 ng/mL). Medium containing 0.5% DMSO served as the non-treated control, while dexamethasone (Dex) at a concentration of 20 µM was used as the positive control. After 24 h of incubation, the supernatants were collected, and IL-6 and IL-8 concentrations were determined using the sandwich ELISA method (BD Biosciences Pharmingen, San Diego, CA, USA).

### 4.7. Inflammatory Activity of HLE

Caco-2 cells were seeded in 48-well plates and cultured as described above. After 48 h of cultivation, HLE was applied to the appropriate wells at concentrations ranging from 3.125 to 100 µg/mL. Medium containing 0.5% DMSO served as the non-treated control, while dexamethasone at a concentration of 20 µM was used as the positive control. Prior to application, HLE was dissolved in DMSO and sterilely transferred into fresh serum-free medium. To remove high-molecular-weight pyrogens, the extract was subjected to a two-step ultrafiltration using a Microsep™ Advance Centrifugal Device with 30 kDa and 100 kDa cut-offs (Pall Corporation, Port Washington, NY, USA), as described in Kruk et al. [50]. After ultrafiltration, 10% FBS was added to the medium. Following a one-hour pre-incubation, IL-1*β* was added to each well to achieve a final concentration of 10 ng/mL. After 24 h of incubation, the supernatants were collected, and IL-6 and IL-8 concentrations were determined using the sandwich ELISA method (BD Biosciences Pharmingen).

### 4.8. TEER Measurement of Caco-2 Monolayer

Caco-2 cells were cultivated in 6-well plates on a Transwell transparent insert (Greiner Bio-One, Kremsmünster, Austria) with density of 6 × 10^5^ cells/insert for 21 days. (37 °C, 5% CO_2_) Medium that was used contained the earlier mentioned ingredients. Three mL of medium was added to the apical compartment, and 3.5 mL was added to the basolateral compartment. The medium was replaced three times per week. CellZscope systems for 24-well and 6-well formats (nanoAnalytics GmbH, Münster, Germany) were used to analyze the TEER. Solutions were made as described in Section 4.5 with concentrations 12.5–100 µg/mL. As non-treated control, medium containing 0.5% DMSO was used. The medium in the apical compartment was replaced with the HLE solutions or control. After a 24-h incubation period 25 ng/mL of IL-1*β* was added in basolateral compartment [20].

### 4.9. Analysis of Caco-2 Transport and Metabolism of HL Constituents

Caco-2 cells were seeded and cultivated as described for the TEER measurements. After 21 days of cultivation, HLE solutions at concentrations of 12.5–100 µg/mL (prepared as for the cell viability assay) were added to the apical compartment. As a non-treated control, medium containing 0.5% DMSO was used. Supernatants from both the apical and basolateral compartments were collected at 0 h and 24 h for chromatographic analysis.

### 4.10. Chromatographic Analysis of Caco-2 Transport and Metabolism of HLE Constituents

Analyses were performed using a Vanquish UHPLC system coupled with an Exploris 120 Orbitrap mass spectrometer (Thermo Scientific, Bremen, Germany) equipped with a Kinetex XB-C18 column (150 × 2.1 mm, 1.7 µm). Mobile phases consisted of (A) water and (B) acetonitrile:water (4:1, *v*/*v*), both containing 0.1% HCOOH and 5 mM NH_4_HCOO. The LC gradient was: 0–3.5 min 1% B, 3.5–16.5 min 1–26% B, 16.5–26.5 min 26–100% B, 26.5–28.5 min 100% B; flow rate 0.3 mL/min; column temperature 45 °C. Electrospray settings: spray voltage 3.5 kV (+)/2 kV (−); sheath/aux/sweep gas 48/11/2 Arb; ion transfer tube 320 °C; vaporizer 280 °C. Data were acquired in full-scan mode (resolution 120,000, polarity switching). MS2 spectra were obtained using stepped NCE 30/50% (resolution 15,000) and ddMS2 settings of four fragmentation scans per full scan, single-charge filter, isotope and dynamic exclusion enabled. Experimental samples were analyzed with and without fragmentation exclusion lists generated from control samples to enhance MS2 coverage. Data processing was performed in Compound Discoverer 3.4 (Thermo Scientific, Austin, TX, USA). Signals corresponding to hop extract constituents were identified based on comparison of donor compartment solution and method blank. Similarity analysis and molecular networking (mzMine 4.6) were based on shared fragments or constant *m*/*z* shifts [33,68].

### 4.11. Statistical Analysis

All statistical analyses were performed using Microsoft Excel 2016 (Microsoft Corp., Redmond, WA, USA), IBM SPSS Statistics 30 (IBM Corp., Armonk, NY, USA), and GraphPad Prism 8 (GraphPad Software, LLC, San Diego, CA, USA). Data are presented as mean ± SD from at least three independent experiments. Statistical significance was assessed using one-way ANOVA followed by Dunnett’s post hoc test, with *p* < 0.05 considered statistically significant.

## 5. Conclusions

Although hop is a well-known medicinal and brewing plant, studies on the biological effects of its complete extracts remain limited. This work presents a comprehensive phytochemical analysis of the full hop extract, its anti-inflammatory activity in an inflammation-induced Caco-2 model, and its selective transport across Caco-2 monolayers. After ultrafiltration used to remove pyrogens, HLE reduced IL-6 secretion in a concentration-dependent manner and decreased IL-8 levels, while also restoring IL-1*β*-induced TEER disruption at higher concentrations. Transport studies revealed selective permeability, favoring medium-sized bitter acid derivatives, with three new metabolites detected exclusively in the basolateral compartment. Overall, these findings demonstrate that the whole hop extract exerts moderate but biologically relevant protective effects, without the need for isolating individual fractions and supports its potential application as a practical ingredient in functional foods or dietary supplements.

## Figures and Tables

**Figure 1 ijms-26-10608-f001:**
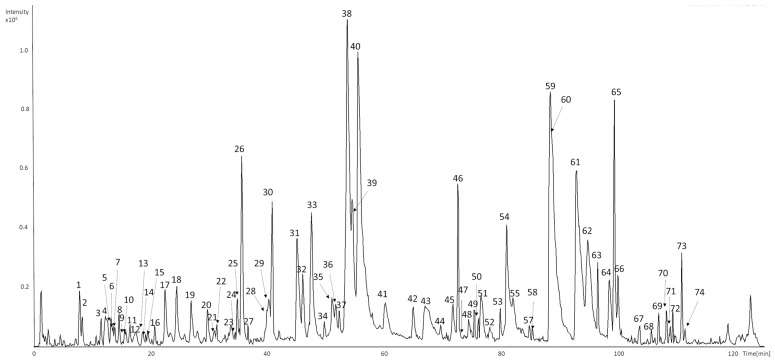
UHPLC-DAD-MS/MS base peak chromatograms (BPC) of representative samples of HLE recorded in negative mode.

**Figure 2 ijms-26-10608-f002:**
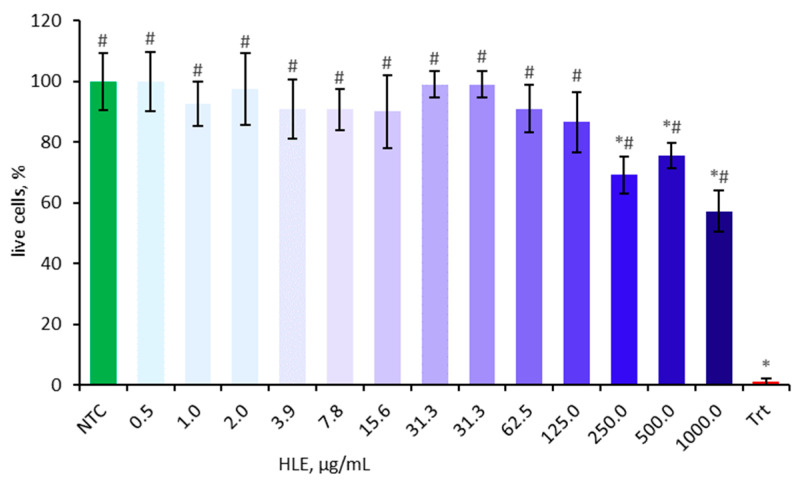
Effect of HLE (0.5–1000 µg/mL) on the viability of Caco-2 cells. Triton-X100 (Trt, 0.1% *v*/*v*) was used as a positive control. Data were expressed as mean ± SD of three separate experiments conducted in triplicate. Statistical significance was determined by Dunnett’s post hoc test at *p* ≤ 0.05 versus NTC (*) or positive control (Trt) (#).

**Figure 3 ijms-26-10608-f003:**
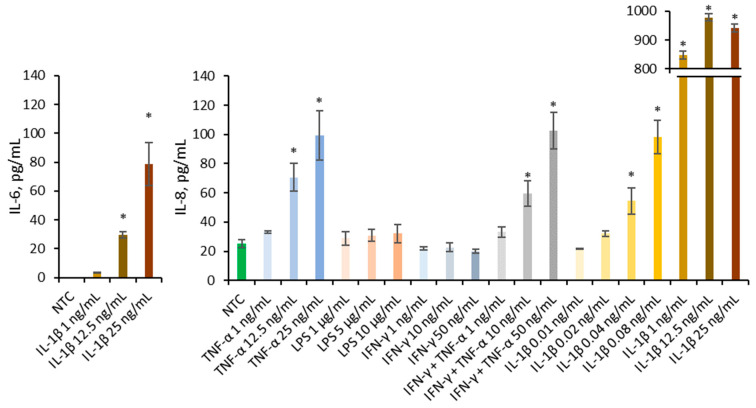
Effect of TNF-α (1, 12.5, 25 ng/mL), LPS (1, 5, 10 µg/mL), IFN-*γ* (1, 10, 50 ng/mL), mixture of IFN-*γ* and TNF-α (1 + 1 ng/mL, 10 + 10 ng/mL, 50 + 50 ng/mL), and IL-1*β* (0.01, 0.02, 0.04, 0.08, 1, 12.5, 25 ng/mL) on the IL-6 and IL-8 production in Caco-2. Data were expressed as mean ± SD of three separate experiments conducted in triplicate. Statistical significance * *p* < 0.05 versus non-treated control (NTC) (Dunnett’s post hoc test).

**Figure 4 ijms-26-10608-f004:**
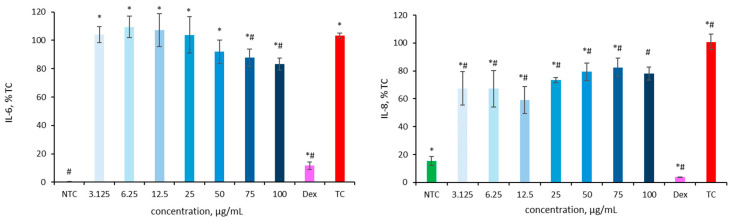
Effect of HLE (3.125–100 µg/mL) on IL-6 and IL-8 production in Caco-2 cells stimulated IL-1*β* (10 ng/mL). Interleukins levels are expressed as a percentage relative to the non-treated control (NTC). As a positive control dexamethasone (Dex, 20 µM) was used. Data are presented as mean ± SD (*n* = 3). Statistical significance was determined by Dunnett’s post hoc test at *p* ≤ 0.05 versus NTC (*) or the IL-1*β*-stimulated control (TC) (#).

**Figure 5 ijms-26-10608-f005:**
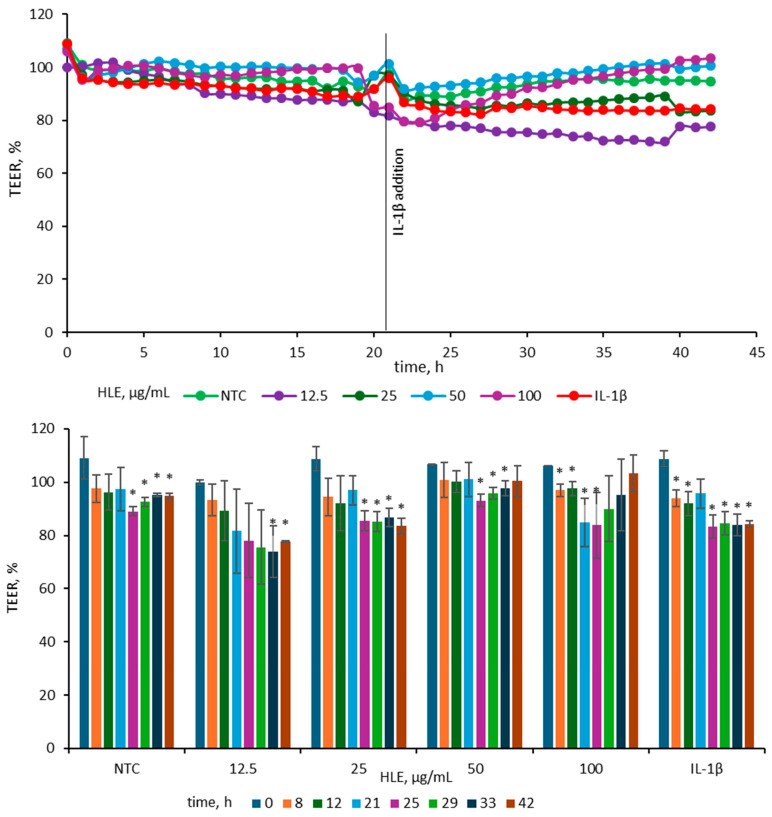
Effect of HLE (12.5–100 µg/mL) on TEER values of Caco-2 monolayers before and after stimulation with IL-1*β* (10 ng/mL). IL-1*β* was added at 21 h, while HLE samples were applied at 0 h. The upper graph depicts mean TEER values (*n* = 3), and the lower graph shows TEER changes at selected time points with standard deviations and statistical significance. Results are expressed as mean ± SD (*n* = 3). Statistical significance (*) was determined using Dunnett’s post hoc test at *p* ≤ 0.05 versus TEER at 0 h for selected samples.

**Figure 6 ijms-26-10608-f006:**
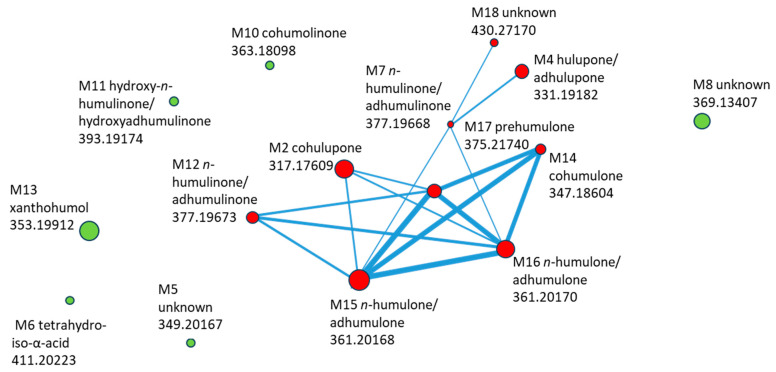
Molecular network of HLE constituents transported through Caco-2 monolayer, with compound ID, identification, and *m*/*z* (negative mode). Compounds marked in red are included in the cluster, while those marked in green are outside the cluster.

**Table 2 ijms-26-10608-t002:** UHPLC-DAD-MS/MS data of HLE constituents and their putative metabolites present in the acceptor compartment after 24 h incubation of the extract with Caco-2 (negative ion mode); b—base peak, FA—formic acid.

Peak ID	*t*_r_, min	*m*/*z*	Calculated *M*_W_	Reference Ion	Identification	MS2	Transport Efficiency (%)
**M1**	16.4	343.13944	298.14139	[M+FA−H]^−^	metabolite 1	55.86654, 58.08405, 135.14658b, 166.25183	n/a
**M2**	22.2	317.17609	318.18336	[M−H]^−^	cohulupone	152.04805, 180.04312, 205.05089, 248.10583b, 317.17654	7.2
**M3**	22.6	449.14987	450.15715	[M−H]^−^	metabolite 2	78.17171, 87.84658, 236.79556b, 385.18146	n/a
**M4**	22.7	331.19182	332.19904	[M−H]^−^	hulupone/adhulupone	166.06361, 194.05853, 219.0666, 262.12131, 331.19196b	3.7
**M5**	22.8	349.20167	350.20907	[M−H]^−^	unknown	125.06069, 221.15474, 305.21255b, 306.21475, 349.20239	2.7
**M6**	22.8	411.20223	412.20951	[M−H]^−^	tetrahydro-iso-α-acid	125.06058, 153.09175, 209.08215b, 263.12863, 297.1351	2
**M7**	23.3	377.19668	378.20396	[M−H]^−^	*n*-humulinone/adhumulinone isomer 1	221.08199, 249.07697, 291.12399, 308.12689 b, 377.19846	6.9
**M8**	23.4	369.13407	370.14135	[M−H]^−^	unknown	119.05026b, 143.45879, 178.45453, 355.06036, 369.13239	7.2
**M9**	23.8	431.18389	432.19117	[M−H]^−^	metabolite 3	208.58945, 321.09552, 329.75766b, 345.68106	n/a
**M10**	23.9	363.18098	364.18826	[M−H]^−^	cohumulinone	195.06659, 223.06175, 224.06908b, 294.11105, 363.18164	9
**M11**	24.2	393.19174	394.199	[M−H]^−^	hydroxy-*n*-humulinone/hydroxyadhumolinone	98.9873, 195.06644b, 207.06679, 375.18094, 393.19287	17.4
**M12**	24.4	377.19673	378.204	[M−H]^−^	*n*-humulinone/adhumulinone isomer 2	237.07733, 238.08496, 249.07758, 291.12433, 308.12717b	11.7
**M13**	24.9	353.13912	354.1464	[M−H]^−^	xanthohumol	79.95727, 96.95999, 119.05023, 353.19995b, 354.20297	3.5
**M14**	26.3	347.18604	348.19332	[M−H]^−^	cohumulone	93.05042, 207.06635, 235.0614, 278.11609b, 279.11969	9.5
**M15**	26.8	361.20168	362.20896	[M−H]^−^	*n*-humulone/adhumulone isomer 1	207.06651b, 221.08208, 249.07739, 292.13202, 293.13544	8
**M16**	27.0	361.20170	362.20898	[M−H]^−^	*n*-humulone/adhumulone isomer 2	207.06624, 221.08202, 249.07722, 292.13193b, 293.13556	9.6
**M17**	27.5	375.21740	376.22468	[M−H]^−^	prehumulone	221.08205, 235.09769, 263.09299, 306.14771b, 307.15103	4.3
**M18**	27.8	429.26446	430.27174	[M−H]^−^	unknown	259.09827, 301.14468, 317.13989, 360.19376, 429.26514b	26.7

## Data Availability

Data will be made available on request.

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
