# Peer review of "Integrated Molecular and Functional Analysis of Hop Ethanolic Extract in Caco-2 Cells: Insights into Inflammation, Barrier Function, and Transport"

_ijms, 2025, doi:10.3390/ijms262110608_

Round 1

Reviewer 1 Report

Comments and Suggestions for Authors

Review report
Journal:  IJMS (MDPI)
Manuscript ID:  ijms-3953256
Title: Integrated molecular and functional analysis of hop ethanolic 2 extract in Caco-2 cells: insights into inflammation, barrier function, and transport.

This manuscript describes the phytochemical, anti-inflammatory and bioavailability properties of a Humulus lupulus extract. As the authors describe, the work provides a comprehensive view on the hop extract as a whole, which is different from most published literature on hops.

Some comments / questions upon reading the manuscript:

- In the work, all studies are carried out on an ethanolic extract (70% ethanol) of Humulus lupulus. Why was this type of extraction solvent chosen? It would be valuable to explain this in brief in the manuscript, as of course this is an important parameter affecting all other outcomes.

- Line 108: “catechin or epicatechin oligomers” ? do these represent procyanidin A-type dimers? For an ion with m/z 575 this would plausible, but an ion with m/z 565 is reported. And what would the tentative structure of the ion with m/z 579 be? In reference 27, Kruk et al. (2024) there is no mentioning of ions with m/z 579 or m/z 565 and apart from the ion at 289 (representing the (epi)catechin monomer ions) the other fragments also are not reported in this reference, so this reference does not seem to fully support the assignment. It’d be useful to add at least one other citation to support these tentative identifications.

- Line 121-123: “Peak 27, with a deprotonated molecular ion at m/z 533, produced MS2 fragment ions at m/z 209, 293, 323, and 413, corresponding to sequential losses of two hexose moieties (−162 Da × 2)”. Could this be further clarified? Losses of 162 Da from m/z 533 results in m/z 371, which is not reported as fragment and non of the reported fragments shows a difference of 162 Da either, so the sequential loss of these two hexose moieties can not be deduced from the m/z data provided.

- Lines 148-150: could you please add the typical values for neutral loss of a prenyl or hydroxyl group? Loss of a hydroxy-group would normally lead to a neutral loss of 16 Da, but looking at the fragments, I see for example fragments 369 en 387, as well as 399 and 417 with a difference of 18 Da, which indicates a dehydration. Could it be clarified if indeed a de-hydroxylation or a dehydration is meant?

- Paragraph 4.7: here is it mentioned that dexamethasone was used as positive control at a concentration of 20 micromolar, while in Figure 1 (which should be Figure 4) 20 millimolar is written. Please check and correct.

Other minor remarks:

- line 48-49: suggestion to change this sentence to: A well-characterized property of these PFs is their immunomodulatory activity, which is consistently confirmed by research.
- Line 106: notation of λ_max → change to λmax
- Line 107, 108, 118: m/z should be written in italics. Please check if this is in order throughout the manuscript.
- Line 116, 121: MS2 or MS2 (2 in superscript)? Be consistent throughout the manuscript.
- Line 127: 31-74 → compound numbers in bold.
- Figure 1: please add the ionization mode to the title of the figure.
- Figure 2: What do the # above the bar charts indicate? Add this to the figure title.
- Line 176: in vitro in italics
- Line 178: [17].Only → add space in between
- Line 178-179: “was able to significant increase IL-6 at a concentration of 25 and 12.5 ng/mL”→ was able to increase IL-6 at concentrations of 25 and 12.5 ng/mL significantly.
- Line 127 and others: for the n-humulinone compounds → n in italics or not? Not consistent throughout the manuscript.
- The figure with graphs of effect of HLE on IL-6 and IL-8 is numbered Figure 1, but this should be Figure 4.
- Line 276: Only a subset… were able → only a subset… was able
- Two tables are numbered as Table 1, so please correct.
In table 1 retention times are shown with one decimal number, while in table 2 with 3 decimal numbers. This seems like an overkill to me and is not consistent, so I’d suggest to only use 1 decimal number here too.
Out of curiosity: how was it deduced that metabolite 1 is a formic acid adduct, while none of the other detected ions is observed as formic acid adduct?
- Line 305: contribute → contribute to
- Line 316: screen → screening
- Line 317: HLE demonstrated dose-dependent cytotoxicity effect → leave out “effect”
- Line 319: equal → equal to
- line 337: change to: attenuate it, thus eliminating
- line 339: change to: where levels of IL-6 and IL-8 were increased
- Lines 349-352: please check this sentence and rephrase, as it doesn’t seem grammatically correct
- Line 355: “(Restivo et al., 2023; Sangiovanni et al., 2019; Vázquez-Cervantes et al., 2021)” Please confirm these correspond to references 12, 15 and 21 and remove.
- Line 364: by contrast → in contrast
- Line 425, 450, 466, 494, 495: ml → mL
- Paragraph 4.2: what amount of plant material was extracted?
- Line 438-444: the flow rate was 0.3 mL/min, but 2 different columns were used. Was the same flow rate applied to both? Why were these two different columns used and not one single column?
- Line 449: caco-2 → Caco-2
- Line 456: MMT → MTT
- Line 456-457: “in a 48-wells plate were Caco-2 cells seeded…” → Caco-2 cells were seeded in a 48-well plate…
- Line 462: “the same in for every well” → this doesn’t seem right, please correct.
- Line 464-465: “solution of medium and MTT (0.5 mg/mL)” → solution of MTT (0.5 mg/mL) in medium
- Line 471: E. coli in italics
- Line 485: Devices → Device
- Line 492: 6 wells plate → 6-well plates
- Line 494: 3 → three (at beginning of sentence)
- Line 498: “in cell viability section” → in section 4.5
- Line 523: recognized → identified or distinguished?
- Line 529: which versions of SPSS and GraphPad Prism were used?

Author Response

Response to Reviewer 

We sincerely thank the reviewer for the insightful and constructive comments, which were extremely helpful, particularly in clarifying and refining the interpretation of the LC-MS data.

1. In the work, all studies are carried out on an ethanolic extract (70% ethanol) of Humulus lupulus. Why was this type of extraction solvent chosen? It would be valuable to explain this in brief in the manuscript, as of course this is an important parameter affecting all other outcomes.

A detailed explanation has been added to Section 4.2. “The use of 70% ethanol provides an optimal balance between polarity and extraction efficiency, allowing for the recovery of both hydrophilic and moderately lipophilic compounds. More polar solvents (e.g., water) or nonpolar organic solvents (e.g., dichloromethane) would extract different fractions of constituents and may not reflect the phytochemical composition relevant to dietary exposure.”

2.  Line 108: “catechin or epicatechin oligomers” ? do these represent procyanidin A-type dimers? For an ion with m/z 575 this would plausible, but an ion with m/z 565 is reported. And what would the tentative structure of the ion with m/z 579 be? In reference 27, Kruk et al. (2024) there is no mentioning of ions with m/z 579 or m/z 565 and apart from the ion at 289 (representing the (epi)catechin monomer ions) the other fragments also are not reported in this reference, so this reference does not seem to fully support the assignment. It’d be useful to add at least one other citation to support these tentative identifications.

We appreciate the reviewer’s careful observation. Compound no. 15 was indeed incorrectly annotated and has been corrected to “unknown,” for which we apologize. Compound 16a was tentatively assigned as a catechin/epicatechin derivative based solely on the presence of the fragment ion at m/z 289, characteristic of catechin-related structures. This correction has been implemented in the revised text, which now reads:

Compound detected at peak 16a exhibited [M−H]⁻ ions at m/z 579, with product ions at m/z 231, 289, and 533. The fragmentation pattern, with catechin/epicatechin-related ion at m/z 289, suggests the presence of a catechin-derived structure [33], although this identification remains tentative.”

3. Line 121-123: “Peak 27, with a deprotonated molecular ion at m/z 533, produced MS2 fragment ion× 2)”. Could this be further clarified? Losses of 162 Da from m/z 533 results in m/z 371, which is not reported as fragment and non of the reported fragments shows a difference of 162 Da either, so the sequential loss of these two hexose moieties can not be deduced from the m/z data provided.

We thank the reviewer for this valuable observation. The description has been clarified in the revised version to avoid ambiguity. The intended interpretation referred to the overall loss of two hexose units (2 × 162 Da) from the precursor ion m/z 533, yielding the fragment at m/z 209, corresponding to the acylphloroglucinol aglycone. Although an intermediate ion corresponding to a single hexose loss (m/z 371) was not detected, the total mass difference between the parent ion and the aglycone fragment (324 Da) supports the proposed dihexoside substitution pattern. The revised text now reads:

Peak 27, with a deprotonated molecular ion at m/z 533 ([M–H]⁻), produced MS² fragment ions at m/z 413, 323, 293, and 209. The fragmentation pattern is consistent with the successive neutral losses of two hexose moieties (−162 Da × 2) from the parent ion, ultimately yielding the aglycone fragment at m/z 209. Although an intermediate ion corresponding to a single hexose loss was not detected, the overall mass difference between m/z 533 and 209 (324 Da) supports the proposed acylphloroglucinol dihexoside structure. The base peak at m/z 209 represents the acylphloroglucinol aglycone, in agreement with previously reported data for acylphloroglucinol derivatives [34,35].”

4. Lines 148-150: could you please add the typical values for neutral loss of a prenyl or hydroxyl group? Loss of a hydroxy-group would normally lead to a neutral loss of 16 Da, but looking at the fragments, I see for example fragments 369 en 387, as well as 399 and 417 with a difference of 18 Da, which indicates a dehydration. Could it be clarified if indeed a de-hydroxylation or a dehydration is meant?

We thank the reviewer for this insightful comment. The initial description was indeed ambiguous. Upon re-evaluation, we found that the reviewer correctly noted the 18 Da difference corresponding to dehydration rather than dehydroxylation. Furthermore, it appears that the reviewer may have overlooked that peaks 71, 73, and 74 all exhibited fragmentation to m/z 399, which is characteristic of bitter acid–type compounds. Peak 70 was re-evaluated and corrected to “unknown.” The revised description now reads:

Late-eluting peaks 71, 73, and 74 with molecular ion at m/z 513 exhibited MS² spectra consistent with dehydration (−18 Da) and characteristic neutral losses of 96 and 114 Da. In addition, fragmentation to m/z 399 was observed, suggesting the formation of a colupulone-type ion. According to the fragmentation behavior reported by Česlová et al. [45], who observed similar fragment ions in low-polarity hop constituents, together with their late retention time and the absence of characteristic UV absorption maxima, these compounds were tentatively assigned as bitter-acid derivatives [45,46].”

5. Paragraph 4.7: here is it mentioned that dexamethasone was used as positive control at a concentration of 20 micromolar, while in Figure 1 (which should be Figure 4) 20 millimolar is written. Please check and correct.

We thank the reviewer for noticing this error. The correct concentration is 20 µM (micromolar), not 20 mM. This has been corrected in both the text (Section 4.7) and Figure 4.

5. Other minor remarks (lines, figures, tables, grammar, and formatting corrections as listed above). In addition: Out of curiosity: how was it deduced that metabolite 1 is a formic acid adduct, while none of the other detected ions is observed as formic acid adduct?

We thank the reviewer for the thorough and careful reading of our manuscript and for all these detailed editorial and technical suggestions. All the indicated corrections — including linguistic improvements, style unification (m/z, in vitro, λmax, italics, consistent use of MS² and compound numbering), grammatical adjustments, clarification of figure and table numbering, correction of units (mL), and specification of extraction yield, flow rate, and software versions — have been implemented in the revised manuscript in accordance with the reviewer’s recommendations. We also carefully verified the entire manuscript to ensure overall consistency and correctness of terminology and formatting.

Regarding metabolite 1, the software identified this feature as a formic acid adduct based on the simultaneous detection of [M–H]⁻ and [M+FA–H]⁻ ions in the MS¹ spectra, with the exact mass difference corresponding to the mass of the formic acid moiety. This automated recognition is a built-in function of the feature extraction algorithm, which distinguishes adduct types based on precise mass differentials between co-eluting ions.

Reviewer 2 Report

Comments and Suggestions for Authors

The manuscript presents a comprehensive study on the biological activities of hop ethanolic extract (HLE) using an integrated approach that combines phytochemical profiling with in vitro biological assays. The study provides valuable insights into the anti-inflammatory and barrier-protective effects of HLE in Caco-2 cells. However, there are several areas where the study could be strengthened to provide a more complete picture of the extract's potential therapeutic applications.

  1. The study demonstrates significant anti-inflammatory and barrier-protective effects of HLE but lacks detailed mechanistic insights into how these effects are achieved. Understanding the underlying molecular pathways would strengthen the study. Can the authors provide more detailed insights into the potential molecular mechanisms through which the hop ethanolic extract (HLE) exerts its anti-inflammatory and barrier-protective effects? Are there specific signaling pathways or cellular targets that the authors believe are modulated by HLE?
  2. The study is limited to in vitro experiments using the Caco-2 cell model. While this model is relevant, in vivo validation is necessary to assess the extract's efficacy in a more biologically complex environment. Given the promising in vitro results, what are the authors' plans for validating these findings in in vivo models? Are there any preliminary in vivo studies currently underway that could support the therapeutic potential of HLE?
  3. The study shows that higher concentrations of HLE (above 250 μg/mL) significantly reduce cell viability. This cytotoxicity could limit the practical application of HLE. How do the authors interpret the cytotoxic effects observed at higher concentrations of HLE, and what are the implications for the practical application of HLE in functional foods or dietary supplements?
  4. The anti-inflammatory effects of HLE were not strictly dose-dependent, showing a non-linear trend. This could indicate complex interactions among the extract's components. Can the authors provide a hypothesis for the non-linear dose-response relationship observed in the anti-inflammatory assays? What might be the underlying reasons for this phenomenon?
  5. The study reveals that only certain compounds in the HLE can cross the intestinal epithelial barrier. The reasons for this selective permeability are not fully explored. Can the authors elaborate on the potential reasons for the selective permeability of HLE constituents across the intestinal epithelial barrier? How might these findings influence the formulation of HLE for optimal bioavailability?
  6. The study does not provide a detailed comparison with previous research on isolated hop-derived compounds. This comparison would highlight the advantages of using the whole extract. How do the authors reconcile the findings from this study with previous research on isolated hop-derived compounds? What are the potential advantages of using the whole extract over isolated fractions in terms of biological activity and therapeutic potential?
  7. The study does not address the potential long-term effects of HLE consumption, which could be important for understanding its therapeutic potential. What are the authors' thoughts on the long-term effects of HLE consumption? Are there any potential chronic effects or benefits that could be explored in future studies?
  8. The study does not discuss the clinical relevance of the findings or the steps needed to translate these results into practical applications. How do the authors envision the clinical relevance of their findings? What steps are needed to translate these in vitro results into practical applications for managing inflammatory bowel disease (IBD) or other inflammatory conditions?
  9. The detection of metabolites in the basolateral compartment suggests biotransformation within the Caco-2 monolayer, but the potential metabolic pathways involved are not explored. Can the authors provide more information on the potential metabolic pathways involved in the biotransformation of HLE constituents within the Caco-2 monolayer? What is the significance of these metabolites in the overall biological activity of HLE?

Author Response

Response to Reviewer

We would like to express our sincere gratitude to the Reviewer for the careful reading of our manuscript and for the constructive, insightful comments that substantially deepened the discussion and improved the overall clarity and scientific rigor of the paper. The suggestions were especially valuable in refining the interpretation of the mechanistic aspects, metabolic transformations, and translational implications of our findings. Below, we provide detailed responses to each point and indicate the corresponding revisions introduced in the manuscript.

1. The study demonstrates significant anti-inflammatory and barrier-protective effects of HLE but lacks detailed mechanistic insights into how these effects are achieved. Understanding the underlying molecular pathways would strengthen the study. Can the authors provide more detailed insights into the potential molecular mechanisms through which the hop ethanolic extract (HLE) exerts its anti-inflammatory and barrier-protective effects? Are there specific signaling pathways or cellular targets that the authors believe are modulated by HLE?

Response:
Explanation was added to Discussion:

“The anti-inflammatory and barrier-protective effects of HLE appear to be mediated by the modulation of interconnected signaling pathways. Hop-derived polyphenols suppress NF-κB, ERK, and AKT activation in intestinal epithelial cells, thereby reducing pro-inflammatory cytokine expression [31]. In our model, IL-1β stimulation induced cytokine release and decreased TEER, consistent with reports that IL-1β disrupts tight junctions via NF-κB/ERK-dependent upregulation of myosin light chain kinase (MLCK) [57]. By inhibiting these pathways, HLE likely prevents MLCK activation and preserves the distribution of zonula occludens-1 and occludin, maintaining epithelial integrity. Moreover, prenylated flavonoids such as xanthohumol have been shown to activate AMP-activated protein kinase (AMPK), which stabilizes tight junction proteins under inflammatory stress [58,59]. These mechanisms together explain the concurrent suppression of cytokine release and maintenance of barrier function observed in HLE-treated Caco-2 monolayers.”

2. The study is limited to in vitro experiments using the Caco-2 cell model. While this model is relevant, in vivo validation is necessary to assess the extract's efficacy in a more biologically complex environment. Given the promising in vitro results, what are the authors' plans for validating these findings in in vivo models? Are there any preliminary in vivo studies currently underway that could support the therapeutic potential of HLE?

Response:
We appreciate the reviewer’s insightful comment emphasizing the importance of in vivo validation. An explanatory note has been added to the Discussion section:

“Considering these observations, it should be noted that although the Caco-2 cell model provides a well-established and physiologically relevant tool for investigating intestinal barrier function, it cannot fully reproduce the complexity of the in vivo intestinal environment, including immune responses, microbiota interactions, and systemic factors. Consequently, in vivo studies would constitute a valuable extension of the present work to confirm the protective and anti-inflammatory potential of hop extract and its metabolites under physiologically relevant conditions.”

In addition, we plan to extend this research to in vivo studies focusing on intestinal inflammation and barrier integrity. These experiments will be designed once the necessary ethical approvals and research funding are obtained.

3. The study shows that higher concentrations of HLE (above 250 μg/mL) significantly reduce cell viability. This cytotoxicity could limit the practical application of HLE. How do the authors interpret the cytotoxic effects observed at higher concentrations of HLE, and what are the implications for the practical application of HLE in functional foods or dietary supplements?

Response:
Explanation was added to Discussion:

“To better understand the nature of this effect, it should be noted that the cytotoxicity observed at concentrations above 250 µg/mL may result from the high content of prenylated flavonoids and bitter acid derivatives, which are lipophilic and can perturb membrane integrity or mitochondrial function at excessive doses [51]. Such effects are common for concentrated hop-derived preparations and likely reflect nonspecific stress rather than targeted bioactivity. Importantly, maintaining extract concentrations below 100 µg/mL effectively prevented cytotoxicity in Caco-2 cells, indicating a narrow but safe biological window. Given the expected dilution and metabolic transformation of hop constituents in the gastrointestinal tract, the in vivo exposure levels are anticipated to be far below cytotoxic thresholds, supporting the potential safety of HLE under physiological conditions. This interpretation is further supported by clinical data showing that long-term supplementation with a hop extract standardized in 8-prenylnaringenin was well tolerated over 48 weeks in postmenopausal women [52], confirming the overall safety of hop-derived preparations during chronic use.”

4. The anti-inflammatory effects of HLE were not strictly dose-dependent, showing a non-linear trend. This could indicate complex interactions among the extract's components. Can the authors provide a hypothesis for the non-linear dose-response relationship observed in the anti-inflammatory assays? What might be the underlying reasons for this phenomenon?

Response:
Explanation was added to Discussion:

“Similar non-linear, biphasic (“hormetic”) responses have been widely reported for polyphenol-rich plant extracts, including those containing prenylated flavonoids. At lower concentrations, such compounds can activate adaptive antioxidant and anti-inflammatory signaling, whereas higher levels may induce mild oxidative or metabolic stress, resulting in partial loss of efficacy [55,56]. Therefore, the non-linear trend observed for HLE likely reflects the balance between synergistic and antagonistic interactions among its diverse constituents, as well as concentration-dependent cellular adaptive responses.”

5. The study reveals that only certain compounds in the HLE can cross the intestinal epithelial barrier. The reasons for this selective permeability are not fully explored. Can the authors elaborate on the potential reasons for the selective permeability of HLE constituents across the intestinal epithelial barrier? How might these findings influence the formulation of HLE for optimal bioavailability?

Response:
Explanation was added to Discussion:

“These metabolic conversions, together with the physicochemical diversity of hop constituents, likely contribute to the selective permeability pattern observed in this study. This pattern is consistent with established structure–permeability relationships for polyphenols, where molecular size, degree of conjugation, and hydrophobicity critically influence transport across the intestinal barrier [63,64]. Bitter-acid derivatives likely diffuse passively through the lipid bilayer, whereas large, highly prenylated flavonoids and flavan-3-ol oligomers exhibit limited permeability due to steric hindrance and affinity for membrane or protein binding sites. Similar findings have been reported for other polyphenol-rich extracts studied in Caco-2 [65].”

6. The study does not provide a detailed comparison with previous research on isolated hop-derived compounds. This comparison would highlight the advantages of using the whole extract. How do the authors reconcile the findings from this study with previous research on isolated hop-derived compounds? What are the potential advantages of using the whole extract over isolated fractions in terms of biological activity and therapeutic potential?

Response:
Explanation was added to Discussion:

“Previous studies have mainly investigated isolated hop-derived compounds such as xanthohumol, isoxanthohumol, and 8-prenylnaringenin, which were shown to suppress NF-κB signaling, reduce cytokine release, and protect tight-junction integrity under inflammatory conditions [15,17]. In contrast, the present study demonstrates that a complete hop ethanolic extract exerts broader and more balanced biological effects, likely due to synergistic interactions among polyphenols, bitter acids, and prenylated chalcones. This compositional complexity may underlie the moderate but consistent anti-inflammatory and barrier-protective responses observed, supporting the concept that whole-plant extracts can modulate multiple cellular pathways more effectively than isolated constituents.”

7. The study does not address the potential long-term effects of HLE consumption, which could be important for understanding its therapeutic potential. What are the authors' thoughts on the long-term effects of HLE consumption? Are there any potential chronic effects or benefits that could be explored in future studies?

Response:
Explanation was added to Discussion:

“This interpretation is further supported by clinical data showing that long-term supplementation with a hop extract standardized in 8-prenylnaringenin was well tolerated over 48 weeks in postmenopausal women [52], confirming the overall safety of hop-derived preparations during chronic use.”

8. The study does not discuss the clinical relevance of the findings or the steps needed to translate these results into practical applications. How do the authors envision the clinical relevance of their findings? What steps are needed to translate these in vitro results into practical applications for managing inflammatory bowel disease (IBD) or other inflammatory conditions?

Response:
Explanation was added to Discussion:

“The findings of this study may have translational relevance for intestinal health. By showing that HLE attenuates IL-1β–induced cytokine release and preserves epithelial barrier integrity, our results highlight its potential as a source of bioactive compounds supporting intestinal homeostasis. While direct extrapolation to clinical outcomes is limited, these effects parallel key mechanisms involved in IBD. Future studies should therefore employ complex in vitro or ex vivo models and in vivo validation to confirm efficacy and safety under physiological conditions.”

9. The detection of metabolites in the basolateral compartment suggests biotransformation within the Caco-2 monolayer, but the potential metabolic pathways involved are not explored. Can the authors provide more information on the potential metabolic pathways involved in the biotransformation of HLE constituents within the Caco-2 monolayer? What is the significance of these metabolites in the overall biological activity of HLE?

Response:
Explanation was added to Discussion:

“The few features detected exclusively basolaterally (putative metabolites M1, M3, and M9) likely represent smaller, more mobile transformation products formed within the monolayer. The detection of unique features in the basolateral compartment suggests partial biotransformation of HLE constituents within the Caco-2 monolayer. No characteristic mass shifts corresponding to glucuronide or sulfate conjugates were detected in MS spectra, indicating that phase II conjugation was unlikely to occur. Instead, the formation of smaller and more polar features suggests oxidative or reductive modifications consistent with limited phase I metabolism reported in Caco-2 cells [60–62]. Such transformations may increase solubility and facilitate transcellular transport while maintaining biological activity [63]. These metabolic conversions, together with the physicochemical diversity of hop constituents, likely contribute to the selective permeability pattern observed in this study.”

Reviewer 3 Report

Comments and Suggestions for Authors

This study“Integrated molecular and functional analysis of hop ethanolic extract in Caco-2 cells: insights into inflammation, barrier function, and transport” presents an integrated chemical and functional analysis of a hop ethanolic extract (HLE). Untargeted HPLC-MS profiling identified 85 compounds, including hydroxychinamic acid derivatives, flavan-3-ols, flavonol glycosides, bitter acid derivatives, and prenylated flavonoids. In an inflammation-induced Caco-2 model, the ultrafiltered HLE exhibited anti-inflammatory activity, as evidenced by the concentration-dependent reduction of IL-6 and decreased IL-8 secretion, alongside barrier-protective effects indicated by the mitigation of IL-1β-induced TEER disruption. Transport assessment across intestinal epithelial monolayers revealed the selective permeability of medium-sized bitter-acid derivatives and suggested metabolic transformation during epithelial passage. The innovative use of the whole extract, without isolating individual fractions, provides a practical and physiologically relevant approach, supporting its potential application in the formulation of functional foods or dietary supplements.However, there are several areas for improvement:

  1. The introduction lacks a systematic review of existing literature. A clearer description of the research gaps concerning the biological activity of *complete* hop extracts, compared to studies on isolated compounds or specific fractions, is necessary.
  2. In section 4.6, 25 ng/mL IL-1β was required to significantly increase IL-6 secretion during model selection. However, subsequent experiments in sections 4.7 (anti-inflammatory activity) and 4.8 (TEER measurement) used 10 ng/mL and 25 ng/mL IL-1β, respectively. Please clarify the rationale for using different concentrations of the inflammatory trigger.
  3. The statistical significance criteria (e.g., the specific p-value threshold, such as p < 0.05) should be explicitly stated in the main text or the figure legends for clarity.
  4. Section 4.2 mentions the extract was "concentrated with a vacuum evaporator and lyophilized." Please specify the detailed conditions for these steps, such as the rotary evaporation temperature and the duration of lyophilization.
  5. What was the extraction yield of the final lyophilized HLE? Please provide the percentage yield.

Author Response

Response to Reviewer 

We would like to sincerely thank the Reviewer for the valuable and detailed comments, which significantly improved the quality and precision of our manuscript. The suggestions helped us clarify methodological aspects and strengthen the rationale of the study. All comments were carefully addressed as detailed below.

1. The introduction lacks a systematic review of existing literature. A clearer description of the research gaps concerning the biological activity of complete hop extracts, compared to studies on isolated compounds or specific fractions, is necessary.

Response:
To Introduction was added the following parts:

“Numerous studies have focused on isolated compounds or defined fractions derived from Humulus lupulus, particularly prenylated flavonoids such as xanthohumol, isoxanthohumol, and 8-prenylnaringenin. XTM has been shown to exert potent antioxidant, anti-inflammatory, and antiproliferative activities in vitro by modulating key signaling pathways including Nuclear Factor kappa-light-chain-enhancer of activated B cells (NF-κB), nuclear factor erythroid 2-related factor 2 (Nrf2), and mitogen-activated protein kinases (MAPKs) [15,17]. In human peripheral blood mononuclear cells, oral intake of XTM attenuated lipoteichoic acid-induced inflammatory responses, supporting its systemic immunomodulatory potential [18]. More recently, xanthohumol administration effectively alleviated dextran sulfate sodium (DSS)-induced colitis and the accompanying bone loss in mice by modulating the gut microbiota and tryptophan metabolism, thereby protecting the intestinal barrier and improving bone structure and density [19]. Although these studies consistently demonstrate biological activity of individual hop-derived compounds, they rely on reductionist approaches that do not reflect the complex phytochemical interactions present in whole hop extracts and may therefore overlook potential synergistic or antagonistic effects relevant to intestinal physiology.”

“Recent studies have shown that Humulus lupulus extracts display notable antioxidant and anti-inflammatory properties in various biological models. For instance, Hurth et al. (2022) demonstrated that a complete hydroalcoholic extract of H. lupulus significantly reduced interleukin-6 (IL-6) and interleukin-8 (IL-8) secretion and inhibited extracellular signal-regulated kinase (ERK) and p38 MAPK signaling in UVB-stimulated keratinocytes, confirming its broad anti-inflammatory potential [29]. Likewise, Hamm et al. reported that oral administration of hop extract to ovariectomized mice (a model of estrogen deficiency), partially prevented visceral and hepatic fat accumulation, without significantly affecting intestinal permeability, inflammatory cytokine expression, or overall gut microbiota composition [30]. Most recently, Caban et al. demonstrated that a spent hop extract and its polysaccharide-based encapsulates alleviated intestinal inflammation by down-regulating NF-κB, ERK, and protein kinase B (AKT) signaling pathways in HIEC-6 and CCD841CoN cell lines [31].”

“Despite these valuable contributions, important research gaps remain. Most studies on whole extracts focus on non-intestinal models or lack systematic phytochemical characterization, and there are no comprehensive analyses linking the full metabolite profile of hop extracts with their anti-inflammatory and barrier-protective effects in IEC models, as well as with cellular transport and metabolism of extract constituents. The interplay and potential synergistic actions among hop acids, prenylated flavonoids, and other phenolics thus remain poorly understood in the intestinal context.”

2. In section 4.6, 25 ng/mL IL-1β was required to significantly increase IL-6 secretion during model selection. However, subsequent experiments in sections 4.7 (anti-inflammatory activity) and 4.8 (TEER measurement) used 10 ng/mL and 25 ng/mL IL-1β, respectively. Please clarify the rationale for using different concentrations of the inflammatory trigger.

Response:
Added to Section 2.5:

“Caco-2 monolayers treated with various concentrations of HLE and subsequently stimulated with IL-1β (25 ng/mL). The concentration of IL-1β (25 ng/mL) was chosen according to previous findings [20], where among several pro-inflammatory agents tested (IL-1β, TNF-α and IFN-γ and LPS), only IL-1β at this concentration induced a marked reduction in TEER in Caco-2 cell monolayers. After 24 hours of exposure to 25 ng/mL IL-1β, the TEER value in our model decreased by 16 ± 1%, which is consistent with the reduction of approximately 20 ± 4% reported by Van de Walle et al. [20].”

3. The statistical significance criteria (e.g., the specific p-value threshold, such as p < 0.05) should be explicitly stated in the main text or the figure legends for clarity.

Response:
Added to Section 4.11:

“Data are presented as mean ± SD from at least three independent experiments. Statistical significance was assessed using one-way ANOVA followed by Dunnett’s post hoc test, with p < 0.05 considered statistically significant.”

This information was also added to the figure legends.

4. Section 4.2 mentions the extract was "concentrated with a vacuum evaporator and lyophilized." Please specify the detailed conditions for these steps, such as the rotary evaporation temperature and the duration of lyophilization.

Response:
Added to Section 4.2:

“After that, the residue was concentrated using a rotary evaporator (240 mbar, 35 °C water bath, flask rotation speed 120 rpm) to remove ethanol, and subsequently lyophilized to remove water (48 h, vacuum <1 mBar, collector temperature -80 °C).”

5. What was the extraction yield of the final lyophilized HLE? Please provide the percentage yield.

Response:
Added to Section 4.2:

“The yield of dry extract after lyophilization was 13.6%.”

Round 2

Reviewer 2 Report

Comments and Suggestions for Authors

It can be accepted in the current version